# Quantitative Ultrasound Techniques Used for Peripheral Nerve Assessment

**DOI:** 10.3390/diagnostics13050956

**Published:** 2023-03-02

**Authors:** Saeed Jerban, Victor Barrère, Michael Andre, Eric Y. Chang, Sameer B. Shah

**Affiliations:** 1Department of Radiology, University of California, San Diego, CA 92093, USA; 2Research Service, Veterans Affairs San Diego Healthcare System, San Diego, CA 92161, USA; 3Department of Orthopaedic Surgery, University of California, San Diego, CA 92093, USA; 4Department of Bioengineering, University of California, San Diego, CA 92093, USA

**Keywords:** peripheral nerve, quantitative ultrasound, B-mode, elastography, shear wave, ultrasound echogenicity

## Abstract

Aim: This review article describes quantitative ultrasound (QUS) techniques and summarizes their strengths and limitations when applied to peripheral nerves. Methods: A systematic review was conducted on publications after 1990 in Google Scholar, Scopus, and PubMed databases. The search terms “peripheral nerve”, “quantitative ultrasound”, and “elastography ultrasound” were used to identify studies related to this investigation. Results: Based on this literature review, QUS investigations performed on peripheral nerves can be categorized into three main groups: (1) B-mode echogenicity measurements, which are affected by a variety of post-processing algorithms applied during image formation and in subsequent B-mode images; (2) ultrasound (US) elastography, which examines tissue stiffness or elasticity through modalities such as strain ultrasonography or shear wave elastography (SWE). With strain ultrasonography, induced tissue strain, caused by internal or external compression stimuli that distort the tissue, is measured by tracking detectable speckles in the B-mode images. In SWE, the propagation speed of shear waves, generated by externally applied mechanical vibrations or internal US “push pulse” stimuli, is measured to estimate tissue elasticity; (3) the characterization of raw backscattered ultrasound radiofrequency (RF) signals, which provide fundamental ultrasonic tissue parameters, such as the acoustic attenuation and backscattered coefficients, that reflect tissue composition and microstructural properties. Conclusions: QUS techniques allow the objective evaluation of peripheral nerves and reduce operator- or system-associated biases that can influence qualitative B-mode imaging. The application of QUS techniques to peripheral nerves, including their strengths and limitations, were described and discussed in this review to enhance clinical translation.

## 1. Introduction

Ultrasound (US) is an increasingly popular modality for imaging peripheral nerves in the clinic, providing important information about nerve microstructure [1,2]. Many peripheral nerves are located superficially and, hence, are easily accessible for US examination. Although neurophysiological assessments are often considered the gold standard for peripheral nerve assessment [3], US-based techniques can be used as a complementary tool, providing anatomical information and localizing lesions more specifically [4], and providing additional data to guide clinical interventions [4,5,6,7]. Using US for peripheral nerve evaluation is advantageous compared to other imaging modalities, particularly in pediatric [4,8] and geriatric [9,10] populations, due to its rapid scanning capability and wide in-clinic availability.

The dominant US imaging technique in medical studies is reflection or pulse-echo technology (echo-ultrasonography), which analyzes the signals returned to the transducer from the macro- and microstructures within and between studied tissues [11,12]. This technique is based on the principle of sonar (i.e., sound navigation and ranging), where the pulse transmitter and pulse receiver are located on the same side of the studied tissues. Sound waves are typically produced by piezoelectric transducers within the US probe that are stimulated by electrical pulses to vibrate or ring at a desired frequency (e.g., 2 to 18 MHz in common clinical probes). The generated sound waves can be focused at a desired depth by using the phased-array technique, which rings individual or subgroups of elements of the transducer array with a specific delayed calculated sequence [11,12].

Four main operational states (modes) of US have been used in medical imaging, including A-, B-, M-, and Doppler mode [11,12]. A-mode, or amplitude modulation mode, is the simplest type of US. A single transducer transmits a sound pressure wave along a line through the skin into the tissue of interest and receives echoes back from interfaces that are encountered. The echoes are plotted on screen as spikes of different amplitude depending on the echo intensity along the line of propagation. The location along the line corresponds to the depth from the probe and is determined by the round-trip time for the echo assuming a value for the speed of sound, commonly 1540 m/s [11,12]. B-mode, or brightness modulation, is the most common form of US imaging; in this mode, a transducer array simultaneously scans a plane through the tissue and displays a two-dimensional (2D) image reconstructed from A-mode data from each transducer [11,12,13]. Echo-ultrasonography’s concept and A-mode data as well as B-mode image generation are presented in Figure 1 in schematics. The brightness of pixels in B-mode images depends on the amplitude of the echoes arising from the depth captured in the corresponding A-mode data. Tissue contrast in B-mode images is obtained based on the differences in the soft tissues’ acoustic properties, including density, sound speed, scattering, and absorption encountered by the propagating sound waves [12]. M-mode, or motion modulation, displays a one-dimensional A-mode line that is displaced horizontally with time. This mode is typically used for measuring the rate and the range of motion in moving body parts, such as imaging the dynamic cardiac chamber walls and valves [14]. The time-displacement graph is generated by continuously measuring the distance of the object in a selected region of interest from its nearest transducer, using the A-mode data of that transducer. Finally, the Doppler mode utilizes the Doppler effect to assess the velocity (direction and speed) of moving structures in the body, most typically blood [14,15]. Using a known transmitted US frequency, echoes received from blood moving toward the transducer are shifted to a higher frequency, while echoes received from blood moving away from the transducer are shifted to a lower frequency. By calculating the frequency shift of a particular sample volume, for example, a jet of blood flow within arteries, its speed and direction can be determined, mapped onto the B-mode image, and thus visualized. This is particularly useful in cardiovascular studies and can be helpful in identifying peripheral nerves and blood vessels. Doppler speed information is quantitative and is often displayed graphically using an overlaid color scheme (directional Doppler) on a co-registered B-mode image.

US devices and transducers are often categorized based on their nominal operating frequencies into the following three ranges: 1–15 MHz is typical for current clinical scanners; 15–30 MHz is typically designated as high-frequency ultrasound (HFUS); and 30–100 MHz is typically designated as ultra-high-frequency ultrasound (UHFUS) [16,17,18]. HFUS and UHFUS developments have resulted in improved image resolution and quality when evaluating superficial tissues in the human body [13,19,20] such as peripheral nerves [21,22], eye [19,23], and skin [13,17,18,20]. Commonly utilized clinical HFUS transducers for nerve imaging operate with peak frequencies of >20 MHz, which produce very high-resolution US images at shallow depths <5 cm. In current clinical practice, peripheral nerves are evaluated qualitatively using grey-scale B-mode images, which provide information about nerve structure, and using color Doppler images to show vascularity [1,5,6,7].

US is a promising modality for assessing the peripheral nerve status, but it is limited by its qualitative basis as well as system- and operator-dependency [12]. Thus, the qualitative interpretation of US images is often variable or inconclusive. For example, neuropathy often leads to altered nerve echogenicity and even the disappearance of fascicular architecture in B-mode images. Increased vascularity, abnormal anatomical structures, and reduced nerve mobility, indicative of tethering, can also be considered in a neuropathy diagnosis with US [5,6,7]. However, such qualitative nerve characterizations may vary for different operators and imaging setups. Morphometric assessments of B-mode images may provide more objective metrics for evaluating nerve structure. For example, the increased cross-sectional area (CSA) of median nerves has often been correlated with the diagnosis of carpal tunnel syndrome (CTS) [5,24]. To account for differences in patient anthropometric characteristics, dimensionless CSA ratios of nerves (e.g., carpal tunnel inlet-to-outlet) have also been used to assess neuropathy [25]. Nevertheless, even these indices may be confounded by operator-induced and platform-specific sources of variability. Moreover, such morphometric measures lack information about the nerves’ microstructure. Figure 2 shows B-mode images on the transverse plane (short axis plane) of the median nerve in an exemplary healthy participant compared with a CTS patient. Remarkably, the fascicles detected in the healthy median nerve cannot be seen in the CTS patient [5].

Quantitative ultrasound (QUS) techniques offer the potential for a more objective evaluation of peripheral nerves, promising the improved diagnosis of neuropathy and nerve injury. QUS approaches have the potential to more reliably reflect differences in the multi-scale composite structure of the nerves (e.g., fascicles, organized bundles of nerve fibers within a fascicle, and axons), which cannot be directly observed by qualitative methods. QUS investigations performed on peripheral nerves can be categorized into three main groups: (1) B-mode echogenicity-based outcomes, (2) elastography, and (3) backscattered RF signal characterization. B-mode echogenicity measures employ post-processing algorithms in B-mode images to determine the visualized microstructure of the tissue. The underlying physical principle of elastography is that tissue stiffness and other tissue mechanical properties can be quantitatively estimated by analyzing the response of the tissue to an applied force. Strain ultrasonography and shear wave elastography (SWE) are the two major classes of US elastography techniques [26]. In strain ultrasonography, tissue strain, as a response to internal or external compression stimuli, is measured by tracking the motion of speckles detectable in B-mode images [26,27]. In SWE, the speed of shear waves generated by external vibrations or tissue compression is measured and correlated to tissue elasticity or stiffness [26,27]. Finally, backscattered RF signal characterization permits the measurement and modeling of fundamental tissue acoustic parameters that relate to composition and microstructure [12,28]. The signals needed to measure these fundamental tissue parameters are often discarded during machine-dependent B-mode image formation.

In this review, we describe the three main categories of QUS techniques and summarize QUS-based investigations performed on peripheral nerves. The focus of this review will be the employed techniques rather than the results of each application case. In addition to summarizing the strengths and the use cases for each approach, this review also describes the potential limitations associated with each QUS technique.

## 2. Materials and Methods

This review conducted between 2020 and 2022 aimed to provide a thorough description of the current QUS techniques including the advantages and limitations. The literature search was performed in PubMed, Scopus, and Google Scholar databases using the following keywords: “peripheral nerve” and “quantitative ultrasound” or “elastography ultrasound” published after the year 1990. The results were first screened automatically and then through title and abstract reading for excluding duplicated records, non-English written reports, non-ultrasound or non-quantitative studies, and review articles. Study selection was performed following the PRISMA 2020 guidelines as summarized in Figure 3. All reviewed QUS-based investigations performed on peripheral nerves are summarized in Table 1.

## 3. B-Mode Echogenicity Measurement

In B-mode images, sonographers typically identify peripheral nerves based on fascicular boundaries; in the transverse plane, nerves appear honeycomb-like, while in the longitudinal plane, nerves display parallel hyperechoic fascicular borders. Figure 4 shows B-mode images in the transverse and longitudinal planes of the median nerve in a healthy volunteer generated with UFH22 (7–10 MHz) and UFH48 (10–22 MHz) US probes on a Vevo MD, FUJIFILM machine. The honeycomb-like structure of the nerves in the transverse plane becomes more obvious in images generated with higher-frequency probes. Nerves differ from tendons, which display a dense fibrillar structure in B-mode [7,69,70], and are additionally distinguished from blood vessels by using Doppler US [5,6,7,68]. B-mode echogenicity carries information about the nerve structure. Specifically, B-mode images comprise both the information of specular reflections at the tissue interfaces (large structures compared to the wavelength), displayed as hyperechoic areas, and the information of the backscattered signal from the microstructure of the tissue (equal to or smaller than the wavelength), displayed as speckle texture in the image.

The measurement of the hypoechoic fractional area of the nerves in US B-mode images, or so-called nerve density index, has been suggested as an objective measurement of overall nerve echogenicity, capable of distinguishing between normal and abnormal nerves. Tagliafico et al. investigated the feasibility of using the peripheral nerve density index for distinguishing between CTS and neurofibromas in the median nerve [29]. Nerve density was found to be significantly higher in patients with CTS and significantly lower in patients with neurofibromas compared with healthy normal subjects. Using an analogous approach, Bignotti et al. concluded that the median nerve density index was significantly lower in patients with limited cutaneous systemic sclerosis compared to healthy controls [22]. Moreover, symptomatic patients demonstrated lower nerve density compared to non-symptomatic patients. Such approaches have been further refined by normalizing nerve echogenicity to its surrounding tissue environment, thus potentially offsetting machine effects and absolute patient-to-patient differences in echogenicity. For example, Byra et al. investigated the nerve–tissue contrast index (NTI), the ratio between the average brightness of the surrounding tissue to the median nerve, in the context of a CTS diagnosis [24]. NTI was found to be significantly higher in CTS patients compared with healthy volunteers. Boom and Visser employed a variety of thresholding methods to measure the hypoechoic fraction (similar to US nerve density) of the ulnar nerve to detect ulnar neuropathy, with a significantly lower average hypoechoic fraction found in a patient group compared with controls [30]. In an attempt to correlate echogenicity measures with more conventional diagnostic approaches, Simon et al. [31] studied the relation between the ulnar nerve hypoechoic fraction and the electrophysiologic “inching” measures [31]. Hypoechoic fraction and CSA were significantly increased in patients with neuropathy immediately distal and proximal to the medial epicondyle. The above-elbow ulnar motor conduction velocity was inversely correlated with both CSA and hypoechoic fraction in limbs with neuropathy. As an example of the potential ambiguity of such approaches, though, asymptomatic and symptomatic limbs of patients demonstrated similar hypoechoic fractions. In addition to these more common echogenicity measures, the texture of US B-mode images may also be evaluated in peripheral nerves, offering a more nuanced assessment of structural heterogeneity within nerves. For example, in cadaveric ulnar nerves, a texture feature index obtained using the gray level co-occurrence matrix algorithm correlated to combined collagen and myelin concentrations obtained from histology [21].

## 4. US Elastography

The concept of US elastography was introduced by Ophir et al. [71], as a generalization of Eisenscher’s echosonography method [72]. Elastography in general can be explained as analogous to a palpation exam performed on the entire tissue volume. During a palpation exam, the physician taps (shears) the tissue with his or her fingers, and qualitatively senses the deformability or stiffness of the examined tissues. Similarly, US elastography estimates tissue elasticity by analyzing the response of the tissue to an applied force monitored with US [26,27]. Strain ultrasonography and SWE are the two major classes of US elastography techniques [26]. In strain ultrasonography, the tissue strain, as a response to internal or external compression stimuli, is measured by tracking the motion of speckles detectable in B-mode images [26,27]. In SWE, the speed of shear waves induced by external vibrations or tissue compression is measured, which is correlated to tissue elasticity or stiffness [26,27]. These basic mechanical parameters are defined and presented schematically in Table 2.

Elasticity and strain are two important concepts in strain elastography and SWE. The elasticity of a material describes its tendency to retain its original size and shape after being subjected to a deforming force or stress. Solids possess shear and volume elasticity such that they resist changes in shape and volume. Liquids, however, possess only volume elasticity such that they resist changes in their volume, but not in their shapes. Soft tissues are comprised of both liquids and solids; therefore, they possess shear and volume elasticities, yet their shear elasticity is significantly lower than their volume elasticity. Strain can be described as the change in the length of the tissue per its unit length (Table 2). The magnitude of the force applied to the unit area, which induces the strain, is known as stress. The modulus of elasticity can be quantified as the ratio of stress to strain (units of N/m^2^, or Pa). It can be quantified depending on the stress and strain direction, such that Young’s modulus (i.e., normal modulus), *E*, quantifies elasticity in the direction normal to the unit volume surface while the shear modulus, *G*, quantifies elasticity in the tangential direction to the unit volume surface. However, when uniform volumetric stress and strain can be assumed (e.g., liquids under pressure), the bulk or volume elastic modulus, *K,* can be described over the unit of volume.

### 4.1. Strain Ultrasonography

In strain ultrasonography, US is employed to measure the induced tissue displacement in the same direction as the applied stress. As shown in Figure 5, the required cyclic mechanical compression can be applied by (a) free-hand cyclic compression (palpation), (b) cardiovascular pulsation or respiratory motion, (c) acoustic radiation force impulse (ARFI), or (d) external mechanical vibration [2]. The induced strain, regardless of the stress application method, can be measured using different approaches depending on the manufacturer: RF echo correlation-based tracking, Doppler processing, or a combination of the two methods [2]. For example, in RF echo correlation-based tracking, RF A-lines are acquired along the axis of displacement; then, their changes in time between different acquisitions allow the measurement of the tissue displacement and the estimation of the normal strain.

In the free-hand compression technique, compression–decompression cycles are performed using the US transducer itself, with force and frequency adjusted by the sonographer to an appropriate range according to a strain indicator on the US screen [34]. It is very important that the compression occurs vertically, without over-compression, and that the tissue does not slip out of the compression plane [33]. Strain ratios (SRs) are often calculated to provide more reproducible indices from strain-ultrasonography-generated images, as the applied stresses cannot be well-controlled [2].

In ARFI-based compression, a long-duration (e.g., 0.1–0.5 ms vs. 0.02 ms pulses in B-mode imaging), high-intensity (e.g., spatial peak pulse average = 1400 W/cm^2^, spatial peak temporal average = 0.7 W/cm^2^) acoustic “pushing pulse” (i.e., ARFI) is used to displace tissue (displacement of ~10–20 μm) perpendicular to the surface. The magnitude of the applied acoustic radiation force, *F*, can be estimated by the acoustic absorption rate in the tissue, *α*, the speed of sound in the tissue, *c*, and the temporal average intensity of the acoustic beam, *I*, according to Equation (1) [2,26,27].
(1)$$F=\frac{2\alpha I}{c}$$

#### Strain Ultrasonography Applied to Peripheral Nerves

Orman et al. [33] investigated the application of free-hand compression in strain ultrasonography to evaluate median nerve mechanical properties. The mean tissue strain was significantly lower in patients with CTS compared to controls, implying higher tissue stiffness. Given that the mean median nerve perimeter and its CSA in patients were also significantly higher than those of controls, the authors concluded that the enlarged nerves were entrapped by the surrounding tissue structures in carpal tunnel. This entrapment created a pre-stressed condition, resulting in increased apparent stiffness; it was hypothesized that freeing the nerve would reduce the apparent stiffness. The reproducibility of the free-hand compression strain ultrasonography technique in detecting CTS was improved by calculating SR against an acoustic coupler rubber (ACR, made of elastic resin) as an external reference (Figure 6, [34]). The higher SR (ACR/nerve) in CTS patients compared with healthy volunteers implied lower nerve strain, similar to earlier studies [33]. Kesikburun et al. [39] later attempted using bone as an internal reference for SR calculation in median nerve evaluation in CTS patients, reporting a four-fold increase in nerve elasticity for the symptomatic wrist versus the asymptomatic wrist. However, considering bone as the reference for SR measurements is challenging, given the differences in sound speed within soft tissues versus bone. Moreover, near-zero values are expected for strains in bone using free-hand compression. In contrast, Nogueira-Barbosa et al. [41] used the flexor digitorum superficialis muscle (i.e., a soft tissue) as an internal reference for SR calculation in the median nerve in patients with chronic leprosy. A significantly lower SR was observed in the leprosy patients compared with the control group. Later, Tezcan et al. [43] used fat as the reference for SR measurement in the median nerve in patients with CTS undergoing low-level laser therapy and reported a significant drop in nerve stiffness (increased strain) after the therapy. Despite the promise of using muscle or fat as internal references, one caveat is that muscle, fat, and nerves may also be affected by a given pathological condition.

Free-hand compression strain ultrasonography was also used to probe neuropathic severity in several studies, including a comparatively subjective colorimetric assessment of strain levels [36] and a more repeatable SR-based approach [37], to distinguish between individuals with and without CTS. For the latter, the outcomes could not distinguish between mild and moderate CTS [37]. However, a comparable SR-based approach (reciprocal of SR) was successfully used to identify individuals with diabetic neuropathy in tibial nerves; here, the outcomes differentiated between neuropathic severity and correlated with morphometric changes [38].

Ambient or passive strain ultrasonography was also deployed to probe neuropathy; however, the measured SR between the median nerve and nearby tendons was insufficient to identify significant differences between CTS and control groups [42]. It is likely that cardiorespiratory pulsations induced inconsistent deformations in the nerve and reference tendons, which resulted in higher variability compared to free-hand compression strain ultrasonography. In contrast, strain ultrasonography with a cyclic compression apparatus improved repeatability by means of a pre-determined cyclic displacement of the transducer (4 mm displacement at 1.5 Hz) [35]. Using this approach, consistent with freehand compression studies, strains were significantly lower while SR and CSA were significantly higher in CTS patients versus controls. In a later study, the same research group [40] added a strain gauge to the cyclic compression apparatus, to measure pressure. The pressure and pressure/strain ratio (a more realistic index to determine elasticity) were both significantly higher in the CTS patients than in controls.

The feasibility of strain ultrasonography using ARFI push pulses was first investigated by Palmeri et al. [32] to visualize peripheral nerves with adequate contrast versus their surrounding tissues (e.g., muscle, fat, and fascia). The purpose was to improve US guidance in monitoring the distribution of injected anesthetic around the targeted nerves. ARFI strain ultrasonography images yielded significant contrast improvements for the distal sciatic nerve structures and brachial plexus peripheral nerves compared with B-mode imaging. B-mode and ARFI image acquisitions were ECG-triggered for in vivo imaging at locations adjacent to arteries. Several ARFI-based strain ultrasonography investigations have also been reported in the literature as parts of SWE-focused studies, which are summarized in the next section. A limitation of ARFI-based approaches is that the intensity of the radiation force is limited by the potential tissue damage by a push at high acoustic power. As a consequence, the magnitude of the displacements as well as the imaging depth are restricted. In the next section, the assessment of shear elastic modulus (G) will be discussed, as it has more recently been preferred for its fast and reliable assessment, and its wider range of measurable values.

### 4.2. Shear Wave Elastography (SWE)

In contrast to strain ultrasonography, which measures physical tissue displacement parallel to the applied normal stress, SWE employs dynamic stress to generate shear waves in perpendicular directions. The measurement of the shear wave speed (SWS) results in an estimation of the tissue’s shear and normal elastic moduli (*G* and *E*) [27]. SWE was developed based on the fact that the shear elastic modulus, *G* (*G* ≈ *E*/3 for semi-incompressible tissues), determines the propagation speed of mechanical waves, SWS, at a magnitude of the square root of *G* divided by tissue density, *ρ*, as presented in Equation (2).
(2)$$\mathrm{SWS}=\sqrt{\frac{G}{\rho }}\approx \sqrt{\frac{E}{3\rho }}$$

Thus, by evaluating SWS, the elastic modulus of a medium can be estimated. SWS remains below 10 m/s on average in soft tissues and can be adequately tracked by B-mode US images [27]. Shear waves propagate faster through stiffer and denser tissues, as well as along the tissue axis aligned with organized fibers (e.g., the long axis in tendons) [73]. As shown in Figure 7, the shear wave generation methods in nerve SWE studies can be categorized mainly as (a) external mechanical vibration, (b) single-point-focused ARFI, and (c) multiple-point-focused ARFI [2].

Although an external vibration device can induce shear waves, as initially proposed by Krouskop et al. [74], most clinical SWE studies utilize ARFI pulses to induce shear waves, as first proposed by Sugimoto et al. in 1990 [75]. ARFI pushing pulses perpendicular to the tissue surface, as noted in Section 3, result in tissue vibration at an ultrasonic frequency and consequent tissue deformation (displacement) within the region of US excitation (ROE), due to sound wave absorption and scattering. The induced deformation generates shear waves that laterally propagate away from the ROE at a much slower velocity (<10 m/s) compared with US pressure pulses (1500 m/s). Although shear wave mode conversion also occurs with conventional B-mode imaging, the force magnitudes are too small to generate tissue motion detectable with conventional US. ARFI pulses can be applied at a single focal location (point shear wave elastography, pSWE) (Figure 7b) or a multi-focal configuration such that each focal zone is interrogated in rapid succession, leading to a cylindrically shaped shear wave extending over a larger depth, enabling real-time shear wave images to be formed (Figure 7c) [2,26,27]. SWE with multi-focal ARFI configuration is called 2D shear wave elastography (2D-SWE) and allows the real-time monitoring of shear waves in 2D for an SWS measurement. Notably, for 2D SWE, shorter propagation distances are utilized due to the limitations in the number of US tracking pulses.

Considering shear wave generation as the first step in SWE, during the second step, US rapid plane wave excitations are used to track tissue displacement while shear waves propagate. In the third step, changes in tissue displacement maps over time are used to calculate SWS. Frame rates for tracking shear waves are typically between 2 kHz and 10 kHz [73]. Scanners may display different versions of a quality index as a measure of confidence in the estimated SWS, which is calculated from the correlation coefficients between frames of the speckle-tracking images. If the frame rate is too low, there is motion of the patient or probe, or there is no well-developed speckle in the region, then the quality index is low.

#### SWE Applied to Peripheral Nerves

ARFI-SWE has been investigated in several studies to detect CTS. The median nerve stiffness and SWS were reported to be significantly higher in CTS patients compared with controls, and often accompanied by an increase in nerve CSA [44,50,52,64]. Stiffness also accurately differentiated between patients with severe CTS and those with mild or moderate [44,50,52,64]. Figure 8 shows the B-mode image, estimated stiffness in kPa, and the shear wave propagation map in the median nerves of a healthy participant (a, b) and a CTS patient (c, d) [52], who demonstrated a higher stiffness and larger CSA. Notably, the shear wave propagation map displays the arrival time of a shear wave using a series of wave contour lines. ARFI-based SWE also revealed similar changes in the median nerve (higher nerve stiffness and CSA) for patients with acromegaly [57].

Several studies have focused on the SWE-based assessment of peripheral nerve neuropathy. Aslan and Analan [54] used SWE to study the effects of chronic flexed wrist posture among chronic stroke patients on median nerve elasticity and CSA. SWS was significantly higher, but CSA was significantly lower (unlike CTS) for the affected side than the unaffected side; however, SWS did not correlate with the elapsed time since stroke. Similarly, SWE revealed significantly greater ulnar nerve stiffness within the cubital tunnel in patients with ulnar neuropathy (i.e., cubital tunnel syndrome) compared to healthy controls [56]; tibial nerve stiffness in patients with diabetic neuropathy compared to non-neuropathic diabetics and healthy controls [48,60,62,65]; and median nerve stiffness in patients with diabetic neuropathy compared to controls [61,62]. Neuropathic changes were often accompanied by increased CSA [60,61,62]; however, some studies have reported a reversed trend in the tibial nerve [65]. Interestingly, though, both the median and tibial nerves were often found to be thinner in diabetic patients who did not display neuropathy, suggestive of nerve atrophy before neuropathy onset [62]. Conversely, other studies have reported higher CSAs in the tibial nerves of patients with diabetic neuropathy [65]. Such controversial findings in CSA changes emphasize the utility of combining QUS and morphometric imaging.

A number of other nerves have also been studied using SWE. SWE and free-hand compression strain ultrasonography in optic nerves of patients with Behcet’s syndrome (a chronic autoimmune disease) [46] and multiple sclerosis [47] showed significantly higher stiffness and lower strains compared with the control group. Optic nerve stiffness was also examined in patients with migraine using SWE and free-hand compression strain ultrasonography; patients displayed an insignificant decrease in calculated strain (stiffness increase), but a significant increase in SWS [63]. In another study, SWE revealed higher elastic moduli in the brachial plexus nerves of patients receiving radiotherapy compared to their contralateral nerves [55]. SWE has also been used to evaluate healthy nerves; for example, sex-dependent differences between elastic moduli in cervical nerve roots have been noted in healthy volunteers [53].

#### Considerations for Evaluating Peripheral Nerves with SWE

Despite its broad utility, the scale of a targeted nerve should be carefully considered when interpreting SWE outcomes. In one study, the mean stiffness of the saphenous nerve was found to be very similar in the short and long axes [66], a finding inconsistent with multiple other investigations showing a higher shear wave speed in the longitudinal plane [1,67]. It is likely that reliable elasticity data could not be acquired in the transverse plane of the nerves [51], potentially due to the similar scale of nerve diameter and the wavelength of the shear waves and a higher signal from wave reflections at the fiber bundles, apparent in the nerve’s transverse plane [1,67]. SWE measures in thinner nerves may also be less accurate as they are spanned by a smaller number of shear wavelengths to be tracked by US. In addition, the speckle tracking is confounded by the fact that the US beam is larger than the nerve, which introduces a partial volume effect that creates additional influences of the surrounding tissues on reported SWS.

Nerve anatomy and tension should also be considered in evaluating SWE outcomes. Median nerves stretched via wrist extension showed a significant increase in SWS (i.e., nerve stiffness) at the wrist compared to mid-forearm [51], possibly due to the (carpal) bone-proximity artifact [49], which occurs when the studied structure is near a rigid plane such as the bone cortex that prevents homogeneous shear wave propagation. Alternatively, increased SWS may reflect nerve tension due to nerve deformation over the extended wrist [76] or increased pressure within the carpal tunnel [77]. Similar phenomena were observed in sciatic nerves, where increased SWS was noted after tensioning through ankle dorsiflexion in combination with knee extension [45], in median nerves with elbow extension [68], and in ulnar nerves with elbow flexion [68].

The correlation between increased tensile loading and increased SWS has also been validated through the ex vivo application of tensile loads [51]. Nerve viability may also influence SWE outcomes; unlike the above study [51], Schrier et al. [67] reported no differences in median nerve elasticity between the median nerve at the wrist and forearm, as measured in cadaveric nerves using SWE and lateral mechanical compression tests. Such differences might be due to the loss of in vivo loading and intra-nerve pressure or post-mortem tissue changes. More recently, Neto et al. [58] investigated the immediate impact of the slump neurodynamics technique, an exercise for the purpose of improving neural mechanosensitivity, on sciatic nerve stiffness using SWE. SWS was measured during passive foot dorsiflexion in a healthy group before and immediately after slump neurodynamics training. Despite significant differences in the sciatic nerve SWS caused by dorsiflexion, SWS did not change after slump neurodynamic loading, suggesting that nerves likely returned to their natural biomechanical state immediately after releasing the tensile load imposed by the slump. The impact of chronic lower-back-related leg pain on sciatic nerve stiffness was investigated during passive foot dorsiflexion by the same research group [59]. The affected limb showed higher sciatic nerve stiffness compared to the unaffected limb of patients; however, no differences were observed between the unaffected limb of patients versus the healthy controls.

## 5. Backscattered Radiofrequency (RF) Signal Characterization

US scanners increasingly permit access to the beam-formed backscattered RF signals through a research interface. Quantifying such raw RF US data allows an assessment of fundamental acoustic tissue parameters, which can be scanner-independent [12,28]. Commonly used RF signal quantifications, such as backscatter coefficient (BSC), attenuation coefficient (AC), structure–function, and stochastic modeling of envelope statistics of backscattered echoes, are highly likely to be related to the tissue composition and microstructure [12,28].

BSC indicates the tissue’s ability to scatter US waves back to the transducer analogous to echogenicity, providing information about tissue structure and composition. Indeed, the backscattered coefficient represents the energy backscattered by the biological tissue as a fraction of the emitted signal. A reliable assessment requires accounting for all the causes of the loss of energy of the emitted wave in addition to the backscatter, such as the quantitative assessment of the tissue AC, but also the diffraction of the transducer and settings on the gain and processing in the scanner. AC indicates the magnitude of the loss of energy (absorption and scattering) during the US wave propagation into and back from tissues. The entropy of the backscattered echo amplitude can also be used as a model-free approach to analyze the backscattered echo statistics [78]. Such model-free approaches in general do not directly provide an estimation of the tissue microstructure, even though statistical correlations can be found between such coefficients and the tissue microstructural parameters.

In addition to model-free approaches to analyze backscattered echo statistics, some model-based methods have been proposed to estimate the microstructural properties of the tissues [12,79]. Such techniques often require solving an inverse equation problem by fitting the accurately measured BSC and AC parameters (functions of the RF spectrum) in a predefined model of the scatterers in the tissue of interest. For example, stochastic models of the backscattered echo statistics provide an indirect estimate of the spatial and size distributions of scattering microstructures within the studied tissue. Envelope statistics can be also modeled using various distribution functions such as Nakagami distribution, which models random distributions of scatterers in the resolution cells (pre-Rayleigh to Rician distributions) [80].

Raw backscattered RF signal characteristics have been used for several soft tissue assessments [12] including liver [81], breast [82], muscle [83], annular pulleys (tendons) of the fingers [84], and skin [85]. Recently, Byra et al. [21] investigated cadaveric ulnar nerves using raw RF signal measures including BSC, AC, Nakagami parameter, and entropy. These specimens were also assessed with histology; the combined collagen/myelin content demonstrated significant correlations with the BSC and entropy.

## 6. QUS Limitations in Nerve Evaluation

Employing any US technique for peripheral nerve assessment is affected by well-described sonography artifacts such as shadowing, insonation angle, reverberation, and clutter artifacts. Moreover, US techniques are affected by variations in the system settings and parameters such as RF frequency, sampling rate, and gains, which occasionally lead to biased results. Since the structures of interest in peripheral nerves are small, higher US frequencies are often preferred for qualitative and quantitative nerve imaging. The higher US frequencies result in a sharp decrease in the signal-to-noise ratio (SNR) as a function of the depth, due to the higher capacity of biological tissues to attenuate high-frequency RF waves. Therefore, the effective imaging depth for quantitative US techniques is often limited to a few centimeters for nerve imaging, which is more challenging for in vivo investigations when considering the variability in the dimensions of the intermediate tissue layers (e.g., skin, muscle, and fat) between the transducer and the nerves.

B-mode echogenicity measures performed on post-processed images (e.g., nerve density) are limited due to their dependence on the scanner settings and manual identification of tissue boundaries [1,2,73].

US elastography-based measures of tissues are limited by assumptions about the tissue’s material behavior (e.g., linear, elastic, isotropic, homogenous, and incompressible material) to simplify the analysis and interpretation of the strain ultrasonography and SWE measurements [2,73]. However, soft tissues are known to be inherently nonlinear, viscoelastic, and heterogeneous [2]. Specifically, including viscosity in the tissue model means that the tissue stiffness and SWS both depend on the excitation ARFI pulse frequency (or frequency of the external mechanical vibrations), which may vary for different US systems and transducers [2]. Moreover, the mechanical nonlinearity in soft tissue’s behavior means that the induced level of strain in response to an ARFI pulse or other external loads depends on the initial strain state of the material; in other words, operator-dependent transducer orientation and initial compression [1,2]. SWE and strain ultrasonography studies of peripheral nerves have recommended single cutoff values for strain, strain ratio, stiffness, and stiffness ratio between different nerve sections, which, in general, showed better sensitivity and specificity levels in neuropathy diagnosis compared with CSA values and their ratio, despite the codependence of CSA and SWS [1]. However, such single cutoff values show large variations depending on the US system and transducer [1].

Backscattered RF signal characterizations often require analyzing the tissue scans relative to a tissue-mimicking calibration phantom, with a known BSC and AC profile in a range of targeting frequencies. These extra calibration scans may be considered highly resource- or time-consuming, and require specialized training. Therefore, backscattered RF signal characterization methods may be especially challenging for patient-oriented in vivo experimental acquisitions. Notably, methods without extra calibration scans have been reported in other tissues (liver and breast) and may serve to guide future peripheral nerve assessments. In such methods, a smaller reference phantom can be scanned concurrently with the tissue and utilized for further BSC corrections. On the other hand, clinical US systems have been shown to have very stable acoustic output and gain, suggesting calibration scans can be stored once and then applied to the subsequent data analysis for extended periods, reducing the impact on clinical throughput. Moreover, the backscattered RF signal characterizations involve more sophisticated data processing, perhaps offline, compared with other QUS categories. As described in Section 5, the model-based backscattered-signal-related measures often require solving an inverse equation problem by fitting the accurately measured BSC and AC parameters in a predefined model of the scatterers in the medium of interest. More sophisticated models may be required for a more accurate assessment of the backscattered RF signal from peripheral nerves by considering their specific microstructure, surrounding tissue, and microenvironment as quantified using immunohistochemical approaches [86]. As is the case for other QUS techniques, raw RF signal characterization can be influenced by the transducer orientation and compression applied by the operator.

## 7. Conclusions

Three main classes of QUS techniques were described and their reported applications on peripheral nerves in the literature were summarized in this study. Neuropathy and nerve injury may affect the composite structure of the nerves at different scales (i.e., fascicles, organized bundles of nerve fibers within a fascicle, and axons), which can be potentially assessed by QUS techniques. Among the discussed QUS techniques, SWE is more appropriate for the mechanical assessment of peripheral nerves while backscattered RF signal characterization is recommended for the microstructural assessment of the nerves. These approaches are important topics for further investigation.

## Figures and Tables

**Figure 1 diagnostics-13-00956-f001:**
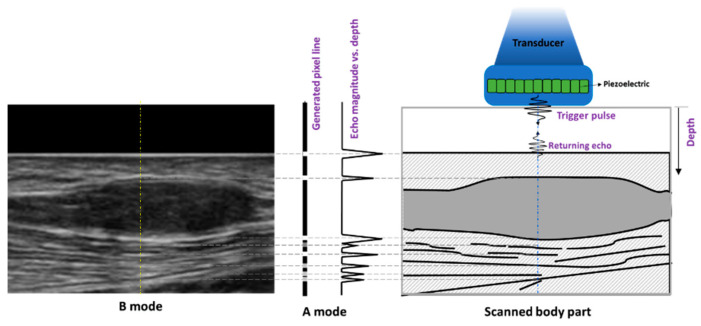
Schematic description of echo-ultrasonography technique as well as A- and B-mode data generation concepts. The returning pulse from the scatterers (schematically shown on the **right**) encountered by the trigger pulse, generated by an exemplary piezoelectric transducer within the US probe, is used to generate A-mode data as well as a pixel line (shown in the **middle**) of the final B-mode image (shown on the **left**).

**Figure 2 diagnostics-13-00956-f002:**
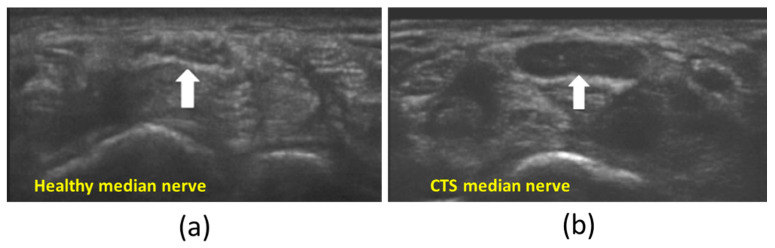
US B-mode images in the transverse plane (short axis plane) of the median nerve (white arrow) in (**a**) a healthy participant and (**b**) a patient with carpal tunnel syndrome (CTS). The fascicles detected in the healthy median nerve cannot be seen in the CTS patient. This figure was previously presented by Vlassakov and Sala-Blanch [5]. Reprinting permission is granted through Rightslink system. Minor modifications were performed for presentation purposes.

**Figure 3 diagnostics-13-00956-f003:**
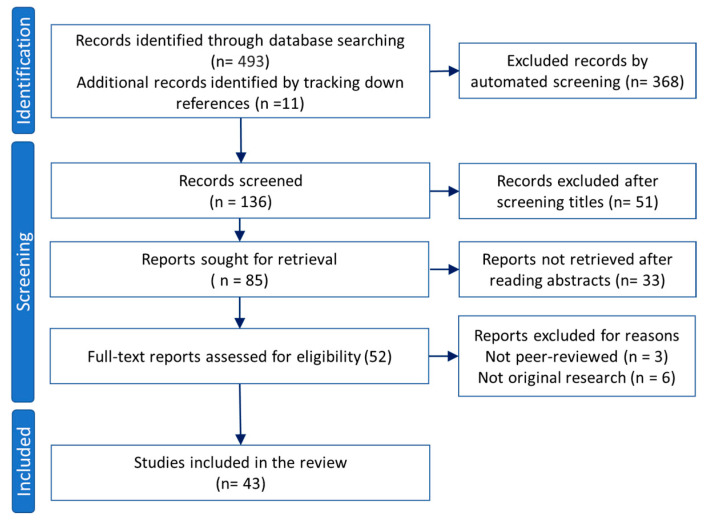
Study selection flowchart based on PRISMA 2020 guidelines.

**Figure 4 diagnostics-13-00956-f004:**
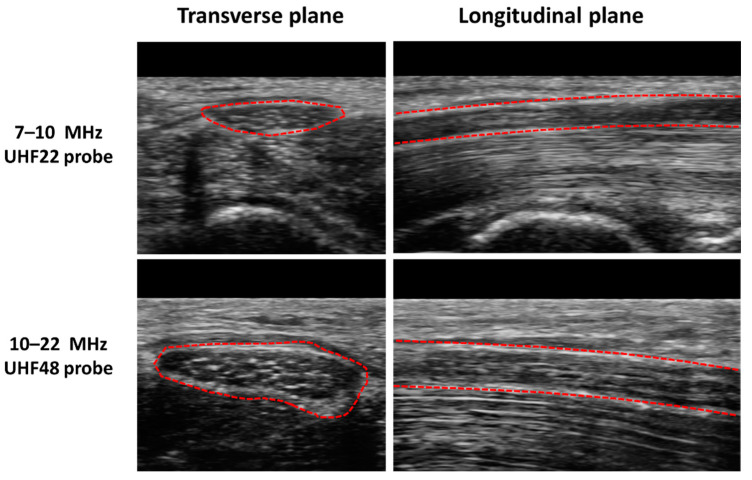
US B-mode images in transverse and longitudinal planes of the median nerve (indicated in red dashed lines) in a healthy volunteer generated with UFH22 (7–10 MHz) and UFH48 (10–22 MHz) US probes on a Vevo MD, FUJIFILM machine. The honeycomb-like structure of the nerves in the transverse plane becomes more obvious in images generated with higher-frequency probes.

**Figure 5 diagnostics-13-00956-f005:**
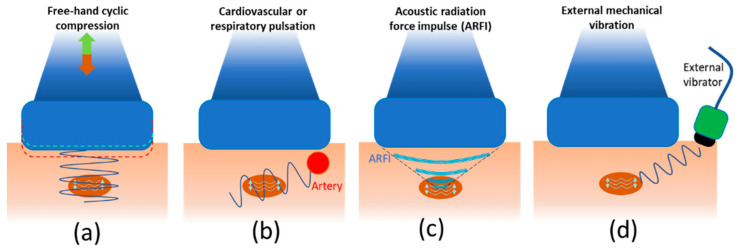
Strain ultrasonography techniques categorized by their excitation methods (generating pressure wave sources) to induce strain in the tissue of interest. (**a**) Free-hand cyclic compression (palpation), (**b**) cardiovascular or respiratory pulsation, (**c**) acoustic radiation force impulse (ARFI), and (**d**) external mechanical vibration have been used as excitation sources in strain ultrasonography.

**Figure 6 diagnostics-13-00956-f006:**
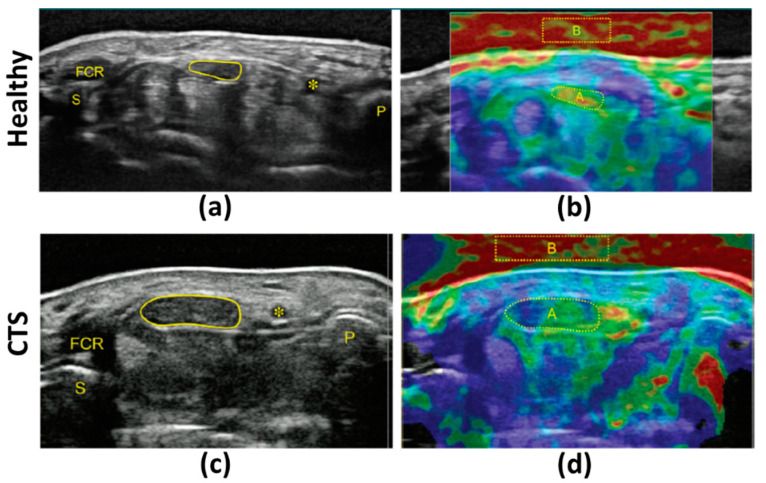
(**a**) B-mode and (**b**) strain images in the transverse plane of the median nerve in a healthy subject (57-year-old female). (**c**) B-mode and (**d**) strain images in the transverse plane of the median nerve in a CTS patient (64-year-old female). Strain ratio (SR) was measured by comparing the strain values in the median nerve (indicated with A in (**b**,**d**)) with an acoustic coupler rubber (ACR) as an external reference (indicated with B in (**b**,**d**)). The median nerve in the CTS patient showed a higher SR (ACR/Median nerve) and CSA. These figures were previously presented by Miyamoto et al. [34]. Reprinting permission is granted by the Radiological Society of North America, which is the copyright holder. Minor modifications were performed for presentation purposes. * = Ulnar artery, FCR = flexor carpi radialis, P = pisiform bone, S = scaphoid bone.

**Figure 7 diagnostics-13-00956-f007:**
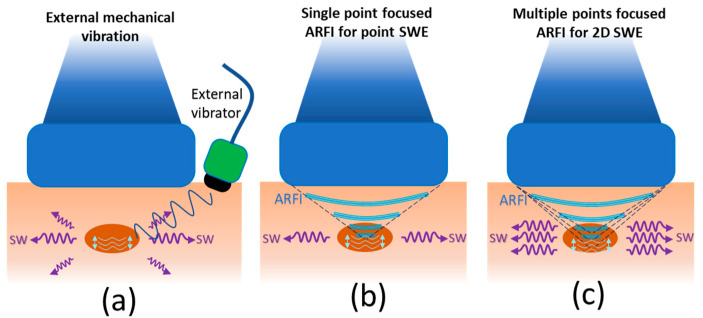
Shear wave elastography (SWE) techniques categorized by their excitation methods to generate shear waves in the tissue of interest: (**a**) external mechanical vibration, (**b**) single-point-focused ARFI for point SWE, and (**c**) multiple-point-focused ARFI for 2D SWE.

**Figure 8 diagnostics-13-00956-f008:**
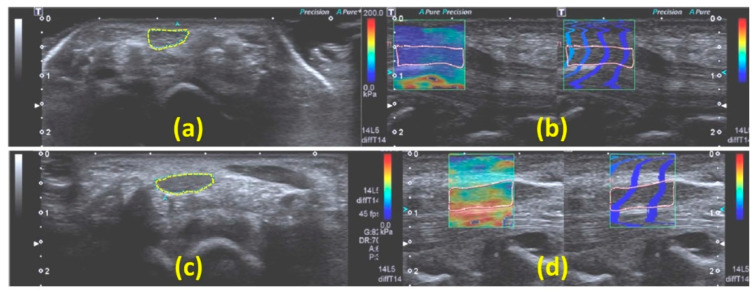
(**a**,**c**) B-mode images of the transverse plane of median nerves of a healthy subject and a CTS patient, respectively. The median nerve cross-section is identified by the dashed line. CSAs of the median nerve were 16 and 27 mm^2^ for the healthy subject and CTS patient, respectively. (**b**,**d**) SWE-based stiffness and propagation maps from SWE on the longitudinal planes of the median nerves of the same healthy subject (**b**) and CTS patient (**d**), respectively. The average stiffnesses measured along the longitudinal plane of the median nerve were 35 and 113 kPa for the healthy subject and CTS patient, respectively. These figures were previously presented by Cingoz et al. [52]. Reprinting permission is granted through Rightslink system. Minor modifications were performed for presentation purposes.

**Table 1 diagnostics-13-00956-t001:** QUS-based investigations performed on peripheral nerves.

Type of QUS	Study	Goal	Anatomy	Scanner	Probe Frequency	Findings
**B-mode echogenicity measurement**	Tagliafico et al., 2010 [29]	To investigate the feasibility of using US-based measure of the peripheral nerve density (hypoechoic fraction) in detecting CTS and neurofibromas.	Median nerve 35 CTS and 30 neurofibroma patients vs. 65 controls Transverse plane	iU-22, Philips (Eindhoven, The Netherlands)	Linear array transducer 5 to 17 MHz frequency range	Nerve density in the median nerve was significantly higher in CTS patients and significantly lower in neurofibroma patients compared with the healthy normal subjects.
	Boom and Visser, 2012 [30]	To investigate the feasibility of using hypoechoic fraction obtained with various thresholding methods in detecting neuropathy.	Ulnar nerve 56 patients with neuropathy vs. 37 controls Transverse plane	Xario XG, Toshiba (Tokyo, Japan)	Linear array transducer (PLT-1204BT) 7 to 18 MHz frequency range	Significantly lower average hypoechoic fraction was found in the patient group compared with the controls using several different thresholding methods.
	Bignotti et al., 2015 [22]	To investigate the feasibility of using US nerve density index in evaluating patients with limited cutaneous systemic sclerosis (lcSSc).	Median nerve 40 lcSSc patients vs. 40 controls Transverse plane	iU-22, Philips (Eindhoven, The Netherlands)	Linear array transducer 5 to 17 MHz frequency range	US nerve density index was found significantly lower in lcSSc patients than in healthy controls. Symptomatic patients demonstrated lower nerve density compared to non-symptomatic patients.
	Simon et al., 2015 [31]	To study the relationship between US B-mode hypoechoic fraction and electrophysiologic measures of the peripheral nerve.	Ulnar nerve in vivo 16 neuropathy patients vs. 52 controls Transverse plane	A Mindray M7 (Shenzen, China)	Linear array transducer (L14-6) 6 to 14 MHz frequency range	Nerve hypoechoic fraction and CSA were significantly increased in patients. Hypoechoic fraction was similar in the asymptomatic and symptomatic limbs of patients. Motor nerve conduction velocity correlated with the maximum hypoechoic fraction and CSA.
	Byra et al., 2020 [21]	To investigate correlations between collagen/myelin content (histology) of nerves with a US-based texture feature, gray level co-occurrence matrix (GLCM).	Ulnar nerve 6 control cadavers, 85 fascicles Transverse plane	Vevo MD, FUJIFILM (Toronto, Canada)	Linear array transducer (UHF48) Center frequency at 30 MHz	GLCM showed a significant correlation with the combined collagen and myelin content of fascicles.
	Byra et al., 2020 [24]	To investigate the feasibility of using the nerve–tissue contrast index (NTI) method in detecting CTS, by considering echogenicity differences in the surrounding tissue.	Median nerve in vivo 10 CTS patients vs. 21 controls Transverse plane	Vevo MD, FUJIFILM (Toronto, Canada)	Linear array transducer (UHF48) Center frequency at 30 MHz	NTI and CSA were significantly higher in patients.
**Strain ultrasonography**	Palmeri et al., 2009 [32]	To investigate nerves’ contrast improvement of strain US vs. B-mode US.	Distal sciatic, brachial plexus, femoral nerves 2 healthy subjects Longitudinal and transverse planes	SONOLINE Antares, Siemens (Erlangen Germany)	Linear array transducer (VF 7-3 and VF I0-5 3 to 7 and 5 to 10 frequency ranges	The strain sonography with ARFI significantly increased the contrast with muscle helping localize nerves for anesthetic injection.
	Orman et al., 2013 [33]	To investigate the potential of strain US (pure strain) free-hand compression in detecting CTS.	Median nerve 41 CTS patients vs. 24 controls Transverse plane	Aplio XG, SSA 790A, Toshiba(Nasushiobara, Japan)	Linear array transducer 12 to 17 MHz frequency ranges	Strain was significantly lower in the patients with CTS. Nerve perimeter and CSA of patients with CTS were significantly higher.
	Miyamoto et al., 2014 [34]	To investigate the capability of free-hand compression strain US technique (strain ratio (SR) with respect to acoustic coupler rubber, ACR) in detecting CTS.	Median nerve 31 CTS patients vs. 22 controls Longitudinal and transverse planes	HI VISION Preirus Hitachi-Aloka Medical (Tokyo, Japan)	Linear array transducer 5 to 18 MHz frequency range AC (Hitachi-Aloka Medical)	Both the strain ratio (AC/nerve) and the CSA in the patients with CTS were significantly higher (stiffer nerves). The presence of CTS was predicted by means of AC/nerve SR and CSA cutoff values of 4.3% and 11 mm2, respectively.
	Yoshii et al., 2015 [35]	To investigate the capability of strain US technique with a cyclic compression apparatus (strain and SR with respect to ACR) in detecting CTS.	Median nerve 8 CTS patients vs. 30 controls Transverse plane	HI VISION Avius, Hitachi Aloka Medical (Tokyo, Japan)	Linear array transducer 5 to 18 MHz frequency range AC (Hitachi-Aloka Medical)	Nerve strains of the patients were significantly lower. Strain ratios, CSA, and perimeters were significantly higher in the patients.
	Ghajarzadeh et al., 2015 [36]	To investigate the capability of free-hand compression strain US technique (blue and red pixel counts in strain image) in detecting CTS severity.	Median nerve 31 CTS patients vs. 22 controls Transverse plane	MYLAB 70 XVG, Esaote (Genoa, Italy)	Linear array transducer 5 to 13 MHz frequency range	Blue indexes in strain images and nerve CSA were significantly different between controls and CTS patients with different levels of disease severity.
	Tatar et al., 2016 [37]	To investigate the capability of free-hand compression strain ultrasonography (SR and strain difference between two nerve sections) in detecting CTS and its severity.	Median nerve 15 mild CTS, 20 moderate CTS patients vs. 18 controls Transverse plane	MYLAB 60, Esaote (Genoa, Italy)	Linear array transducer 4 to 13 MHz frequency range	CTS groups showed significantly stiffer nerves compared with control group. Despite SR-related indexes, CSA-based measures were significantly different between mild and moderate CTS.
**Strain ultrasonography—Continued**	Ishibashi et al., 2016 [38]	To investigate the capability of free-hand compression strain US (SR ratio nerve/ACR) in detecting neuropathy in type 2 diabetes patients.	Tibial nerve 198 type II diabetic patients vs. 29 controls Transverse plane	HI VISION Ascendus, Hitachi Medical (Tokyo, Japan)	Linear array transducer Center frequency at 18 MHz AC (Hitachi-Aloka Medical)	SR in patients without neuropathy was lower compared with controls, further decreasing after developing neuropathy. The tibial nerve CSA in diabetic patients was larger, and increased significantly relative to neuropathy severity. Greater performance was shown for SR versus CSA in detecting nerve neuropathy.
	Kesikburun et al., 2016 [39]	To investigate the capability of free-hand strain US technique (using SR) in detecting CTS.	Median nerve 1 CTS patient (case study) Longitudinal plane	GE LOGIQ S7, GE Healthcare (Yizhuang, China)	Linear array transducer 5 to 12 MHz frequency range	Four-fold increase in nerve elasticity was reported in the symptomatic nerve compared with the healthy wrist.
	Yoshii et al., 2017 [40]	To investigate the capability of strain US technique with a cyclic compression apparatus (strain, AC/nerve SR, strain/applied pressure ratio) in detecting CTS	Median nerve 35 CTS patients vs. 15 controls. Transverse plane	HI VISION Avius, Hitachi Aloka Medical (Tokyo, Japan)	Linear array transducer 5 to 18 MHz frequency range AC (Hitachi-Aloka Medical)	Nerve strain was significantly lower in patients. Pressure and pressure/strain ratio were significantly higher in patients. The ROC curve analyses showed that pressure/strain ratio slightly improved the CTS detection compared with using strain alone
	Nogueira-Barbosa et al., 2017 [41]	To investigate the feasibility of employing the free-hand strain US technique (using SR) in detecting leprosy.	Median nerve 18 leprosy patients vs. 26 controls Transverse and longitudinal planes	SonixRP, Ultrasonix (Richmond, Canada)	Linear array transducer (L14-5) 5 to 14 MHz frequency range	Significantly lower SR was observed for the leprosy patients compared with controls. Leprosy patients with reactions showed lower SR compared with patients without reactions.
	Martin and Cartwright, 2017 [42]	To investigate the feasibility of employing the ambient strain US technique (SR was used) in detecting CTS.	Median nerve 17 CTS patients vs. 26 controls Transverse and longitudinal planes	iU-22, Philips (Eindhoven, The Netherlands)	Linear array transducer 5 to 12 MHz frequency range	Despite the previous literature, no significant differences were found in SR between CTS patients and controls.
	Tezcan et al., 2019 [43]	To investigate the feasibility of employing the strain US technique (SR) in evaluating low-level laser therapy on CTS patients.	Median nerve 34 CTS patients with therapy vs. 17 patients without therapy Transverse plane	ACUSON S3000, Siemens (Erlangen, Germany)	Linear array transducer 4 to 9 MHz frequency range	Mean SR, CSA, and clinical severity scores (SSS, and FSS) decreased significantly after laser therapy.
**Shear wave elastography (SWE)**	Kantarci et al., 2014 [44]	To investigate the potential of SWE (stiffness) with ARFI push pulses in detecting CTS.	Median nerve 37 CTS patients vs. 18 controls Longitudinal plane	Aixplorer; SuperSonic Imagine (Les Jardins de la Duranne, France)	Linear array transducer 4 to 15 MHz frequency range	Nerve stiffness was significantly higher in the CTS group compared with controls. Stiffness was significantly higher in the severe CTS group compared with the mild or moderate severity group.
	Andrade et al., 2016 [45]	To employ SWE (shear wave velocity, SWV) to detect the changes in sciatic nerve stiffness during human ankle motion.	Sciatic nerve 9 healthy volunteers Longitudinal plane	Aixplorer; SuperSonic Imagine (Les Jardins de la Duranne, France)	Linear array transducer (SL 10–2) 2 to 10 MHz frequency range	SWV in the sciatic nerve significantly increased during dorsiflexion when the knee was extended (knee 180°), but no changes were observed for the knee at 90°. SWV in the nerve decreased non-significantly after five ankle dorsiflexions.
	Inal et al., 2017 [46]	To investigate the feasibility of optic nerve evaluations with free-hand compression strain US and SWE in Behcet’s patients.	Optic nerve 46 Behcet’s patients vs. 54 controls Longitudinal plane	LOGIQ E9, GE Healthcare (Wauwatosa, USA)	Linear array transducer 6 to 9 MHz frequency range	Significantly higher stiffness and lower strain were observed in the optic nerve of Behcet’s patients compared with healthy volunteers.
	Inal et al., 2017 [47]	To investigate the feasibility of optic nerve evaluations with free-hand compression strain US and SWE (stiffness) in patients with MS.	Optic nerve in vivo 54 MS patients vs. 59 controls Longitudinal plane	LOGIQ E9, GE Healthcare (Wauwatosa, USA)	Linear array transducer 6 to 9 MHz frequency range	Significantly higher stiffness and lower strain were observed in the optic nerve in MS patients compared with healthy volunteers.
	Dikici et al., 2017 [48]	To investigate the feasibility of using SWE (stiffness) for the diagnosis of diabetic peripheral neuropathy.	Tibial nerve 20 diabetic patients with neuropathy, 20 without neuropathy, and 20 controls Longitudinal plane	Aixplorer; SuperSonic Imagine (Les Jardins de la Duranne, France)	Linear array transducer 4 to 15 MHz frequency range	Diabetic patients without neuropathy had significantly higher stiffness and CSA values compared with control subjects. Patients with neuropathy had much higher stiffness and CSA compared with patients without neuropathy and controls.
	Bortolotto et al., 2017 [49]	To investigate the “bone-proximity” hardening artifacts affecting SWE.	Median nerve 36 healthy volunteers Transverse plane	Aplio 500, Toshiba (Tokyo, Japan)	Linear array transducer 14 MHz frequency	Higher stiffness was reported in nerve sections near bone, and carpel tunnel, compared with the mid-arm.
	Arslan et al., 2018 [50]	To examine the efficiency of SWE (stiffness) in CTS detection and determining CTS severity level.	Median nerve 19 severe, 38 moderate, and 39 mild CTS patient vs. 21 controls Longitudinal plane	ACUSON S2000, Siemens (Mountain View, USA)	Linear array transducer 4 to 9 MHz frequency range	Severe CTS groups showed significantly higher stiffness than mild or moderate severity group. The CSA also showed significant increasing pattern by the severity of CTS.
	Zhu et al., 2018 [51]	To examine the nerve tension impacts on the SWE results (SWS) at different nerve sections.	Median nerve 40 healthy volunteers Longitudinal plane	Aixplorer; SuperSonic Imagine (Les Jardins de la Duranne, France)	Linear array transducer 4 to 15 MHz frequency range	Stretching nerves resulted in a significant increase in SWS. The SWS at wrist was significantly higher than the SWS at the midarm.
	Cingoz et al., 2018 [52]	To compare the SWE evaluation of nerves with MRI diffusion.	Median nerve 35 mild, 9 moderate, 15 severe CTS wrists vs. 18 controls Longitudinal plane	Aplio 500, Toshiba (Tokyo, Japan)	Linear array transducer 14 MHz frequency	CTS patients showed higher stiffness than healthy subjects. Patients with moderate–severe CTS had higher stiffness than patients with mild CTS.
	Bedewi et al., 2018 [53]	To investigate the feasibility of SWE (stiffness) in evaluating the brachial plexus nerves.	Brachial plexus root nerves 40 healthy volunteers Transverse plane	Aixplorer; SuperSonic Imagine (Les Jardins de la Duranne, France)	Linear array transducer 4 to 15 MHz frequency range	Significant differences were found in C6 and C7 nerves between male and female participants. Significant inverse correlation with height was noted at the C6 nerve root.
	Aslan and Analan, 2018 [54]	To study the effects of chronic flexed wrist posture among chronic stroke patients using SWE (SWS).	Median nerve of 24 chronic stroke patients Longitudinal plane	ACUSON S2000, Siemens (Erlangen, Germany	Linear array transducer 4 to 9 MHz frequency range	SWS on the affected side was significantly higher than on the unaffected side. CSA on the affected side was significantly lower than that of the unaffected side. The time elapsed since the stroke showed a significant correlation with CSA.
	Kültür et al., 2018 [55]	To evaluate the brachial plexus after radiotherapy for breast cancer using SWE (stiffness).	Brachial plexus 23 patients underwent radiotherapy. Transverse plane	LOGIQ E9, GE (Waukesha, USA)	Linear array transducer 6 to 9 MHz frequency range	Significantly higher stiffness was estimated for brachial plexuses receiving radiotherapy compared with the contralateral side.
**Shear wave elastography (SWE)—Continued**	Paluch et al., 2018 [56]	To investigate the feasibility of the ulnar neuropathy diagnosis using SWE (stiffness).	Ulnar nerve 34 patients with neuropathy vs. 38 healthy controls Longitudinal axis	iAplio 900, Canon (Tokyo, Japan)	Linear array transducer 5 to 18 MHz frequency range	Patients with ulnar neuropathy presented significantly greater ulnar nerve stiffness in the cubital tunnel and the cubital tunnel to distal and mid-arm stiffness ratio compared with controls. Mean CSA of the ulnar nerve in the cubital tunnel was significantly larger in patients with neuropathy than in controls.
	Burulday et al., 2018 [57]	To evaluate the median nerve of patients with acromegaly using SWE.	Median nerves 15 CTS patients vs. 20 controls Transverse and longitudinal planes	LOGIQ E9, GE (Waukesha, USA)	Linear array transducer 6 to 15 MHz frequency range	Median nerve stiffness and CSA were significantly higher in the patients with acromegaly compared with the control group for both axial and longitudinal nerve planes
	Neto et al., 2019 [58]	To investigate the immediate impact of slump neurodynamics technique on sciatic nerve stiffness using SWE.	Sciatic nerve 14 healthy controls Longitudinal plane	Aixplorer; SuperSonic Imagine (Les Jardins de la Duranne, France)	Linear array transducer (SL 10–2) 2 to 10 MHz frequency range	The sciatic nerve stiffness of healthy participants did not change immediately after a slump neurodynamic technique
	Tiago Neto et al., 2019 [59].	To investigate the impact of chronic lower-back-related pain in legs on sciatic nerve stiffness using SWE.	Sciatic nerve 8 patients with lower-back related pain in legs vs. 8 healthy controls Longitudinal plane	Aixplorer; SuperSonic Imagine (Les Jardins de la Duranne, France)	Linear array transducer (SL 10–2) 2 to 10 MHz frequency range	The affected limb showed higher sciatic nerve stiffness compared to the unaffected limb of the patients. No differences were observed between the unaffected limb of patients and the healthy controls.
	Jiang et al., 2019 [60]	To investigate the feasibility of using SWE (stiffness) for the diagnosis of diabetic peripheral neuropathy (DPN).	Tibial nerve 70 diabetic patients with DPN and without DPN vs. 20 healthy controls Longitudinal plane	Aixplorer; SuperSonic Imagine (Les Jardins de la Duranne, France)	Linear array transducer 4 to 15 MHz frequency range	The tibial nerve stiffness was found to be significantly higher in patients with DPN than that in patients without DPN and control subjects.
	He et al., 2019 [61]	To evaluate patients with diabetic peripheral neuropathy using SWE (stiffness).	Tibial/median nerves 40 diabetic patients with and 40 without peripheral neuropathy vs. 40 controls Longitudinal plane	Aixplorer; SuperSonic Imagine (Les Jardins de la Duranne, France).	Linear array transducer 4 to 15 MHz frequency range	The diabetic patients with neuropathy showed significantly higher nerve stiffness and CSA No significant difference in nerve stiffness was found between the nerves in the left and right limbs in patients.
**Shear wave elastography (SWE)—Continued**	Aslan et al., 2019 [62]	To evaluate adolescent patients with type-I diabetic without peripheral neuropathy using SWE (stiffness).	Tibial and median nerves 25 diabetic patients vs. 32 healthy controls Transverse and Longitudinal planes	Resona 7, Mindray (Shenzhen, China)	Linear array transducer (ComboWave) 4 to 15 MHz frequency range	Both the median nerve and posterior tibial nerve were smaller, and stiffer in the patient group.
	Şahan et al., 2019 [63]	To investigate the feasibility of optic nerve evaluations with free-hand compression strain US and SWE (stiffness) in patients with migraine.	Optic nerve in vivo 30 patients with migraine (16 with and 14 without visual auras) vs. 30 controls. Longitudinal plane	LOGIQ E9, GE Healthcare (Wauwatosa, USA)	Linear array transducer 6 to 15-MHz frequency range	Stiffness from SWE was significantly higher in the optic nerve in patients with migraine compared with controls. A positive correlation was reported between the duration of the disease and the shear modulus.
	Moran et al., 2020 [64]	To examine the efficiency of SWE (stiffness) compared with CSA changes in CTS detection and determining CTS severity level.	Median nerve in vivo 8 severe, 35 moderate, and 36 mild CTS patients vs. and negative EDT controls Longitudinal plane	Aplio 500, Toshiba (Tokyo, Japan)	Linear array transducer 5 to14 MHz frequency	Stiffness and CSA increased according to the CTS severity level. Stiffness was not different between patients with negative and mild CTS findings.
	Wei and Ye, 2020 [65]	To evaluate patients with diabetic peripheral neuropathy using SWE (stiffness).	Tibial nerve 14 diabetic patients with peripheral neuropathy, 13 diabetic patients without peripheral neuropathy vs. 20 healthy controls Longitudinal plane	ACUSON S2000, Siemens (Erlangen, Germany)	Linear array transducer 4 to 9 MHz frequency range	Tibial nerve stiffness in patients with neuropathy and without neuropathy were significantly higher than controls. CSA did not show any significant differences. No significant difference in nerve stiffness was found between the patient with neuropathy and those without neuropathy.
**Shear wave elastography (SWE)—Continued**	Bedewi et al., 2020 [66]	To investigate the feasibility of SWE (stiffness) to evaluate the saphenous nerves.	Saphenous nerves of 36 healthy subjects Transverse and Longitudinal plane	Aixplorer, SuperSonic Imagine (Les Jardins de la Duranne, France)	Linear array transducer 5 to 18 MHz frequency range	Stiffness of the saphenous nerve was found to be very similar in the short and long axes. No correlations of SEW results were found with age, height, weight, and BMI.
	Schrier et al., 2020 [67]	To examine the SWE sensitivity to increasing tensile loading on cadaver nerves at different nerve sections.	Median nerves 10 normal cadaveric wrists Transverse and Longitudinal plane	LOGIQ E9, GE (Waukesha, USA)	Linear array transducer (9LD) 2 to 8 MHz frequency range	SWE- and indentation-based nerve stiffness increased significantly with tensile loading. Acquisition in a transverse plane showed lower values compared with the longitudinal plane. Stiffness did not change when measured proximal to the carpal tunnel.
	Rugel et al., 2020 [68]	To study the limb position impact on the SWE (SWS).	Median and ulnar nerves 16 healthy controls Longitudinal plane	Aixplorer, SuperSonic Imagine (Les Jardins de la Duranne, France)	Linear array transducer 4 to 15 MHz frequency range	SWS increased for limb positions that induced greater tension on the nerves. SWS in median nerve increased by elbow extension. SWS in ulnar nerve increased by elbow flexion.
**Backscattered radiofrequency (RF) signal characterization**	Byra et al., 2019 [21]	To investigate correlations between collagen/myelin content (histology) of nerves with back backscatter coefficient (BSC), attenuation coefficient (AC), Nakagami parameter, and entropy.	Ulnar nerve 6 control cadavers, 85 fascicles Transverse plane	Vevo MD, FUJIFILM (Toronto, Canada)	Linear array transducer (UHF48) Center frequency at 30 MHz	∙ BSC and entropy showed significant correlations with the combined collagen and myelin.

**Table 2 diagnostics-13-00956-t002:** Definition of the basic mechanical parameters described in this study.

Mechanical Parameters	Definition	Formula	Schematics
**Normal stress**	The magnitude of the force applied to the unit area perpendicular to the force direction	$\sigma =\frac{F}{A}$	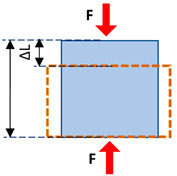
**Normal strain**	The change in length of the tissue per its unit length parallel to the force direction	$\varepsilon =\frac{∆L}{L_{0}}$	
**Young’s modulus (i.e., normal modulus)**	Elasticity defined in applied force direction but perpendicular to the unit volume surface	$E=\frac{\sigma }{\varepsilon }$	
**Shear stress**	The magnitude of the force applied to the unit area parallel to the force direction	$\tau =\frac{F}{A}$	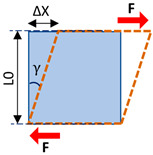
**Shear strain**	The angular change in originally right angles of the unit volume after shear stress application	$\gamma =\frac{∆X}{L_{0}}$	
**Shear modulus**	Elasticity defined in applied force direction and parallel to the unit volume surface	$G=\frac{\tau }{\gamma }$ $G=\frac{E}{2\left(1+v\right)}$	
**Bulk modulus**	Elasticity defined over the unit of volume when a uniform volumetric stress is applied, and a uniform strain is induced	$K=\frac{E}{3\left(1-2v\right)}$	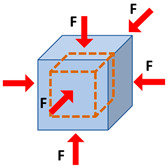

## Data Availability

All data used for this review study are publicly available.

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
