# Peer review of "Quantitative Ultrasound Techniques Used for Peripheral Nerve Assessment"

_diagnostics, 2023, doi:10.3390/diagnostics13050956_

Round 1

Author Response

Reviewer 1

The authors have submitted a very well documented and thorough review of the literature on quantitative ultrasound techniques and their use in assessment of human peripheral nerves. The authors are to be commended for providing clear description of physics, along with helpful diagrams, of the four operational states of imaging ultrasound used in clinical work. The information in the manuscript would be useful to clinicians, as well as to researchers in the field.

Response: We greatly appreciate the support

There are a few issues to be resolved, as detailed below.

  1. The manuscript would benefit from editing for grammar. Examples needing editing are on page 1, line 12, “limitations of when performed”; page 1, line 23, “Third, raw backscattered…” needs a verb; page 1, line 27, “nerves and reducing the operator-…”, replace the “and” with a comma or change “reducing” to “reduce”.

Response: We appreciate the comment and suggestion. The new version of the manuscript has been proofread by native English-speaking editors.

  1. There are a large number of abbreviations used throughout the manuscript. This reviewer suggests including in the manuscript an alphabetized list of abbreviations.

Response: We appreciate the comment and suggestion. A list of abbreviations has been added to the manuscript.

  1. There is a font change on page 1, lines 40-44, which may be a cut and paste error.

Response: We appreciate the comment and suggestion. The font change has been fixed in the new version.

  1. Page 2, line 87, suggests that Doppler mode is “essential” in identifying peripheral nerves and blood vessels. It is certainly helpful; however, this reviewer would argue that Doppler is not essential. Nerves and vessels can often be identified using B-mode, in conjunction with one’s detailed knowledge of anatomy.

Response: We appreciate the comment. The word “essential” has been removed from the sentence and replaced with “can be helpful".

.

  1. Page 3, Figure 1, the image appears to be flipped in orientation. The description indicates that the schematic is shown on the left and the B-mode image on the right. The Figure, however, shows the schematic on the right and B-mode image on the left.

Response: We appreciate the comment. The typo has been fixed in the new version of the manuscript.

  1. Pages 5-9, Table 1, crosses several pages. It is clear where column 1 changes from Bmode to Strain ultrasonography on page 5, and where Shear wave elastography changes to Backscattered radiofrequency on page 9. However, because the column 1 descriptor is “centered”, crossing several rows and pages, it is not possible to discern where Strain ultrasonography changes to Shear wave elastography. This reviewer suggests using the full descriptor in the first row and an abbreviation for each other row in each of the table sections.

Response: We appreciate the comment and suggestion. Table 1 has been modified as suggested.

  1. Pages 5-9, Table 1, there are a few entries that do not include publication year (refs 15, 17, 59). It would also be helpful to list the studies in each section in chronological order.

Response: We appreciate the comment and suggestion. Table 1 has been modified as suggested.

  1. Page 14, line 337, change “branchial” plexus to “brachial” plexus.

Response: We appreciate the comment. The typo has been fixed in the new version of the manuscript.

  1. Page 15, Figure 7, is very small and difficult to review in print. In an online view, the figure must be enlarged several-fold to see the “dashed line” described in the figure legend. Line 404 should read, “transverse plane”. Most importantly, on line 408, the SWE-based maps are mislabeled in the legend. The legend indicates that healthy subject is (a) and CTS patient is (b). However, these should read (b) and (d).

Response: We appreciate the comment and the suggestion. The figure caption has been modified as suggested. The referred dashed lines have been replaced with more visible dashed lines.

  1. Page 16, line 457, includes a reference “PMID:8986667”, which should be replaced with a proper reference.

Response: We appreciate the comment. The proper reference has been added.

  1. Page 17, line 485, includes the abbreviation “SF” for “structure function”; however, the abbreviation is not used subsequently and thus may be removed.

Response: We appreciate the suggestion. SF and other unused abbreviations have been removed from the new version of the manuscript. Those abbreviations that were retained are additionally summarized in the new abbreviation sub-section.

  1. Page 18, lines 555-557, this sentence is unclear: “Notedly, methods without reference phantoms have been reported in other tissues, which can be remarkable to be examined on the peripheral nerve assessments”.

Response: We appreciate the comment. This sentence refers to methods when a separate reference phantom scan is not required, however, a smaller reference phantom can be scanned concurrently with the tissue. This sentence has been modified as follows:

Notably, methods without extra calibration scans have been reported in other tissues (liver and breast), and may serve to guide future peripheral nerve assessments. In such methods, a smaller reference phantom can be scanned concurrently with the tissue and utilized for further BSC corrections. On the other hand, clinical US systems have been shown to have very stable acoustic output and gain, suggesting calibration scans can be stored once and then applied to the subsequent data analysis for extended periods, reducing the impact on clinical throughput.

Reviewer 2 Report

Work well written and complete of information about the topic

Author Response

Response: We greatly appreciate the support.

Reviewer 3 Report

I have no specific comments or suggestions to make regarding this nice and well-written manuscript. The authors provided a wide review of the current literature and knowledge about the use of quantitative USG methods to assess the peripheral nerve. This topic is important in clinical practice as neuroimaging studies related to the peripheral nerve system represent a great development both for diagnosis and follow-up and natural history studies. Most of the key topics were included in the review manuscript and the Methods were also properly disclosed. The only point which should be evaluated by the authors is related to verify the copyright and license to use prior images which were highlighted in this review manuscript. 

Author Response

Response: We greatly appreciate the support. Please note that all required permissions have been granted through Rightslink system (Figure 2 and 8) and RSNA (Figure 6).

Reviewer 4 Report

Reviewer comments

Journal Diagnostics

Title   Quantitative Ultrasound Techniques Used for Peripheral Nerves Assessment

Indeed ,this is an interesting topic and the manuscript covers the US evaluation of nerves in details .

Some points need to be addressed:

12.. strengths and limitations of…. English editing and grammar revision are required in the whole manuscript.

36.. information about nerve microstructure and function… It is not clear how the US gives information about function of the nerves.

42.. used as a complementary tool providing…. this is repetition of the same idea of the previous sentence.

58.. ultrasound. The abbreviations need revaluation. the meaning should be mentioned at the first time and then the abbreviation used later on

58-88...too much details on US technique and only few references

Table 1…there is mis arrangement of the bullets.

Table 1.too long and occupy 3 pages ..it will be better to summarize this table

212..it will be better to add an image illustrating this modality

Strain ultrasonography applied on peripheral nerves and SWE applied on peripheral nerves …are subtitles from the previous titles.

439.. Considerations for evaluating peripheral nerves with SWE…this is also a subtitle. the authors should revise the titles and subtitles arrangement.

576.. are important foci…” foci”…this is an inappropriate word here

Author Response

Reviewer 4:

Indeed ,this is an interesting topic and the manuscript covers the US evaluation of nerves in details .

Response: We greatly appreciate the support

Some points need to be addressed:

  1. . strengths and limitations of…. English editing and grammar revision are required in the whole manuscript.

Response: We appreciate the comment and suggestion. The new version of the manuscript has been proofread by native English-speaking editors.

  1. . information about nerve microstructure and function… It is not clear how the US gives information about function of the nerves.

Response: We appreciate the comment. We agree that the information about nerve function cannot be acquired directly using US. The word “function” has been removed from the sentences.

  1. . used as a complementary tool providing…. this is repetitionof the same idea of the previous sentence.

Response: We appreciate the comment. The redundant sentence has been removed from the manuscript.

  1. . ultrasound. The abbreviations need revaluation. the meaning should be mentioned at the first time and then the abbreviation used later on

Response: We appreciate the comment. Abbreviations (particularly US for ultrasound) were reevaluated and used wherever possible throughout the manuscript. We have also added an abbreviations sub-section, per the additional related request from Reviewer #1.

  1. 58-88...too much details on US technique and only few references

Response: We appreciate the comment. More references have been added wherever required in the referred section.

  1. Table 1…there is mis arrangement of the bullets.

Response: We appreciate the comment. Table 1 has been modified.

  1. Table 1.too long and occupy 3 pages ..it will be better to summarize this table

Response: We appreciate the comment and suggestion. Table 1 was already summarized; however it has been shortened slightly. The column titles have been repeated at each page to facilitate reading through the table. The referred studies have been rearranged in chronological order.

  1. .it will be better to add an image illustrating this modality

Response: We appreciate the comment and suggestion. Schematic illustration of the US elastography techniques have been presented in two separate figures; Figure 4 and 6 illustrates strain imaging and shear wave elastography methods, respectively.

  1. Strain ultrasonography applied on peripheral nerves and SWE applied on peripheral nerves …are subtitles from the previous titles.

Response: We appreciate the suggestion. The subtitles are rearranged as suggested in the new version of the manuscript.

  1. . Considerations for evaluating peripheral nerves with SWE…this is also a subtitle. the authors should revise the titles and subtitles arrangement.

Response: We appreciate the suggestion. The subtitles are rearranged as suggested in the new version of the manuscript.

  1. . are important foci…” foci”…this is an inappropriate word here

Response: We appreciate the comment. The word “foci” has been modified to “topics [for further investigation]”.

Round 2

Reviewer 4 Report

Dear author,

The manuscript improved to a great extent 

Author Response

Editor comments

The authors performed a comprehensive review on peripheral nerves assessment by means of ultrasonography. The manuscript is well-written, and data clearly presented.

Response: We greatly appreciate the support.

The authors are suggested to better characterise ultrasound frequency ranges in order to differentiate high frequency, very high frequency and ultra-high frequency ultrasonographic techniques (see https://doi.org/10.1177/0846537120940684).

Response: We greatly appreciate the suggestion. The typical frequency ranges for HFUS and UHFUS have been added as follows:

US devices and transducers are often categorized based on their nominal operating frequencies into the following three ranges: 1-15 MHz is typical for current clinical scanners; 15-30 MHz is typically designated as high-frequency ultrasound (HFUS); and 30-100 MHz is typically designated as ultra-high-frequency ultrasound (UHFUS) [16–18].

Moreover, in lines 88-90, the authors should mention the application of ultrasonography also in dermatology. Please check https://doi.org/10.1111/jdv.16583 and https://doi.org/10.1177/1534734620972815

Response: We greatly appreciate the suggestion.The application of HFUS and UHFUS in dermatology and ophthalmology have been added as follows to the same paragraph:

 HFUS and UHFUS developments have resulted in improved image resolution and quality when evaluating superficial tissues in the human body [19,20] such as peripheral nerves [21,22], eye [23], and skin [17,18].